# Inactivation of SARS-CoV-2 through Treatment with the Mouth Rinsing Solutions ViruProX^®^ and BacterX^®^ Pro

**DOI:** 10.3390/microorganisms9030521

**Published:** 2021-03-03

**Authors:** Julia Koch-Heier, Helen Hoffmann, Michael Schindler, Adrian Lussi, Oliver Planz

**Affiliations:** 1Interfaculty Institute for Cell Biology, Department of Immunology, Eberhard Karls University of Tuebingen, 72076 Tuebingen, Germany; julia.koch-heier@student.uni-tuebingen.de (J.K.-H.); helen.hoffmann@uni-tuebingen.de (H.H.); 2Institute for Medical Virology and Epidemiology of Viral Disease, Department of Molecular Virology, 72076 Tuebingen, Germany; Michael.Schindler@med.uni-tuebingen.de; 3Department of Operative Dentistry and Periodontology, University Medical Centre, 79106 Freiburg, Germany; adrian.lussi@zmk.unibe.ch; 4School of Dental Medicine, University of Bern, 3010 Bern, Switzerland

**Keywords:** SARS-CoV-2, mouth rinsing solution, ViruProX^®^, BacterX^®^ pro, cetylpyridinium chloride

## Abstract

The *severe acute respiratory syndrome coronavirus 2* (SARS-CoV-2) pandemic effects daily dental work. Therefore, infection control measures are necessary to prevent infection of dental personnel during dental treatments. The use of a preprocedural mouth rinse with chlorhexidine (CHX), cetylpyridinium chloride (CPC), or hydrogen peroxide (H_2_O_2_) solution for 30–60 s may reduce the viral load and may protect the personnel in a dental practice. In the present study the virucidal effect of the mouth rinsing solutions ViruProX^®^ with 0.05% CPC and 1.5% H_2_O_2_ and BacterX^®^ pro containing 0.1% CHX, 0.05% CPC, and 0.005% sodium fluoride (F^-^) was investigated in vitro. The mouth rinsing solutions successfully inactivated infectious SARS-CoV-2 particles, the causative agent of coronavirus disease 2019 (COVID-19), within 30 s. To determine the effective components, CHX, CPC, H_2_O_2_, and a combination of CHX and CPC, were tested against SARS-CoV-2 in addition. While a combination of CPC and CHX as well as CPC alone led to a significant reduction of infectious viral particles, H_2_O_2_ and CHX alone had no virucidal effect against SARS-CoV-2. It can be assumed that preprocedural rinsing of the mouth with ViruProX^®^ or BacterX^®^ pro will reduce the viral load in the oral cavity and could thus lower the transmission of SARS-CoV-2 in dental practice.

## 1. Introduction

*Severe acute respiratory syndrome coronavirus 2* (SARS-CoV-2) is the causative agent of coronavirus disease 2019 (COVID-19) and has become a pandemic with more than 111 million confirmed cases and nearly 2.5 million deaths worldwide (as of February 2021), making it a significant threat to global health (WHO Coronavirus Disease (COVID-19) Dashboard) [1]. Because of the high viral load in the oral cavity, transmission of SARS-CoV-2 from infected individuals can occur through simple processes such as breathing, talking, coughing, or sneezing, and has led to extraordinary challenges for infection control procedures in dental offices [2,3,4,5].

Due to close physical contact, as well as aerosol formation during dental treatment, dentists and their staff are at increased risk for infection and may be a source of further transmission [6,7]. Therefore, specific infection control measures have been implemented in dental offices, including personal protective equipment, and the use of preprocedural mouthwashes to reduce intraoral viral load [7,8,9]. Antiseptic mouthwashes often contain substances (e.g., ethanol, chlorhexidine (CHX), cetylpyridinium chloride (CPC), povidone-iodine (PI), hydrogen peroxide (H_2_O_2_)), which have an antimicrobial activity or fungicidal effect. They can damage or destroy membranes, and some have been proven effective against several viruses [8,9,10]. The broad-spectrum antiseptic CHX showed activity in vitro against lipid-enveloped viruses such as *influenza A virus*, *parainfluenza virus*, *herpes simplex virus 1*, and *hepatitis B virus* but not against nonenveloped viruses [11]. Mouth rinsing solutions often contain CHX but its efficacy against SARS-CoV-2 is only moderate and remains controversial [12,13,14]. Hydrogen peroxide, which is also commonly used in mouth rinsing solutions, might be effective against SARS-CoV-2 since some viruses have been shown to be sensitive to H_2_O_2_. Among others it has been demonstrated to be effective against *adenoviruses*, *rhinovirus*, *influenza A virus*, and *coronavirus strain E229* [15,16,17]. CPC, a quaternary ammonium compound, which is found in some mouth rinsing solutions, showed virucidal activity against *influenza A virus* [18,19]. It inactivates the virus by destroying the capsid [10]. Another study demonstrated efficacy of CPC against HCoV-NL63 and severe CoV (MERS-CoV) [20].

To date, there is little scientific evidence to recommend mouthwashes with antiviral effects against SARS-CoV-2 to reduce viral load in the oral cavity and until now it is not known which mouth rinsing solutions or which components are effective against this novel *coronavirus*. Official recommendations for dental practice therefore give little guidance on specific compounds for preprocedural mouth rinses. Recommendations by the World Health Organization (WHO) include the use of 1% H_2_O_2_ and 0.2% PI solutions. The Center for Disease Control and Prevention (CDC) lists CHX, essential oils, PI, and CPC to be used in preprocedural mouth rinses [21,22]. Therefore, the aim of our study was to investigate two more mouth rinsing solutions: ViruProX^®^, which contains CPC and H_2_O_2_, and BacterX^®^ pro, containing CHX, CPC, and sodium fluoride, for their virucidal activity against SARS-CoV-2 in vitro. Furthermore, the individual components of these mouth rinsing solutions were tested to find out which has antiviral properties against SARS-CoV-2. According to the manufacturer ViruProX^®^ and BacterX^®^ pro are usually applied pure to rinse the oral cavity for 20–40 s. To evaluate if this procedure is sufficient to successfully inactivate SARS-CoV-2, in the following study infectious SARS-CoV-2 particles were incubated with the two solutions and some of their individual components for 30 s and analyzed for their virucidal efficacy. As the null hypothesis of the study it was assumed that none of the tested solutions would have a virucidal activity against SARS-CoV-2 and therefore will show no significant difference compared to the medium control. This hypothesis should be rejected in the present study.

## 2. Materials and Methods

### 2.1. Virus and Cells

*Severe acute respiratory syndrome coronavirus 2* (SARS-CoV-2; Isolate “FI-100”) was provided by the Westfaelische Wilhelms-University Muenster Institute of Virology (IVM) and passaged three times on Vero E6 cells. Titer: 1.25 × 10^7^ plaque forming units per milliliter (pfu/mL). Vero E6 cells, *Cercopithecus aethiops* kidney epithelial cells were obtained from the American Type Culture Collection (ATCC). Cells were grown at 37 °C in Iscove Modified Dulbecco Media (IMDM) (Thermo Fisher Scientific, Waltham, MA, USA) supplemented with 10% fetal bovine serum (FBS) (Capricorn Scientific GmbH, Ebsdorfergrund, Germany) and 1% Penicillin-Streptomycin (Sigma Aldrich, St. Louis, MO, USA).

### 2.2. Test Solutions

The mouth rinses ViruProX^®^ (0.05% CPC and 1.5% H_2_O_2_) (Dr. Wittmann GmbH & Co KG, Zwingenberg, Germany) and BacterX^®^ pro (0.1% CHX, 0.05% CPC, and 0.005% F^−^, without ethanol) (Dr. Wittmann GmbH & Co KG, Zwingenberg, Germany) (Appendix A) and some of their individual components as 0.05% CPC solution, 0.1% CHX solution, a combination of 0.05% CPC with 0.1% CHX solution, and 1.5% H_2_O_2_ solution were used in this study (Appendix A).

### 2.3. Virucidal Activity Test

Infectious SARS-CoV-2 particles (1 × 10^6^ pfu) in 80 µL infection medium (IMDM (Thermo Fisher Scientific), 5% FCS (Capricorn Scientific GmbH), 1% Penicillin-Streptomycin (Sigma Aldrich)) were diluted 1:2 with the mouth rinsing solutions ViruProX^®^ or BacterX^®^ pro or components thereof. In parallel a nonvirucidal medium control of SARS-CoV-2 with infection medium (1 × 10^6^ pfu in 80 µL infection medium + 80 µL infection medium) and a no-virus control (80 µL infection medium + 80 µL test solution) were prepared accordingly. The samples were mixed briefly and incubated at 37 °C for 30 s. Immediately after, 1440 µL infection medium were added leading to a 1:20 dilution of either virus or test solution. Afterwards, the sample was diluted in infection medium up to 1:40,000, which was analyzed in technical duplicates in a plaque assay to allow for separated, single plaques per well and to avoid cytotoxic effects due to the test solutions.

### 2.4. Avicel Plaque Assay

Vero E6 cells (1 × 10^6^ cells/well) were seeded in 6-well plates and incubated for 24 h at 37 °C, 5% CO_2_ in a humidified incubator. The next day the cells were washed once with infection medium before inoculation with 500 µL/well of the prediluted virucidal activity test samples for 1 h at 37 °C, 5% CO_2_ in a humidified incubator. The virucidal activity test samples were defined as the “mouth rinsing solutions” (ViruProX^®^ or BacterX^®^ pro or their individual components incubated with SARS-CoV-2 particles), the “medium control” (SARS-CoV-2 particles + infection medium) and the “no-virus control” (mouth rinse solution + infection medium). The no-virus control was prepared to rule out potential cytopathic effects by the test solutions. After incubation, the inoculum was removed completely, and the cells were overlaid with Avicel-Medium (1.25% Avicel (FMC Biopolymer Germany GmbH, Hamburg, Germany), 10% Minimum Essential Media (MEM) (Thermo Fisher Scientific), 0.01% DEAE-Dextran (Sigma Aldrich), 2.8% NaHCO_3_ (Merck, Darmstadt, Germany), 1% Penicillin-Streptomycin (Sigma Aldrich), 0.2% BSA (Carl Roth, Karlsruhe, Germany), 1% L-Glutamine (Sigma Aldrich)) for 72 h at 37 °C, 5% CO_2_ in a humidified incubator. The Avicel-Medium was removed, the cells were rinsed twice with PBS (Thermo Fisher Scientific) and fixed with 4% Roti-Histofix (Carl Roth) in PBS (Lonza, Basel, Switzerland) for 30 min at 4 °C before staining with crystal violet solution (1% crystal violet (Merck), 10% ethanol in ddH_2_O). Plaques were counted to determine the virus titer in the sample compared to the medium control. Statistical analysis was performed using unpaired t-test (*p* < 0.05). Furthermore, the crystal violet staining was dissolved with methanol and the absorbance was measured at 595 nm, for quantification of the cytopathic effect. Statistical analysis was performed using one-way ANOVA with Tukey’s multiple comparisons test (*p* < 0.05) using GraphPad Prism version 9.0.1 (GraphPad Software, LLC, San Diego, CA, USA).

## 3. Results

### 3.1. Virucidal Activity of Mouth Rinsing Solutions Against SARS-CoV-2

To investigate the virucidal activity of the mouth rinsing solutions ViruProX^®^ and BacterX^®^ pro against the novel *severe acute respiratory syndrome coronavirus 2* (SARS-CoV-2), the mouth rinsing solutions were incubated for 30 s with 1 × 10^6^ plaque forming units (pfu) of SARS-CoV-2 strain FI-100. The remaining infectivity was tested by a standard plaque assay using Vero E6 cells. To avoid a cytotoxic effect of ViruProX^®^ and BacterX^®^ pro and their components on Vero E6 cells, the test solution/virus mixture was diluted 1:40,000. This also resulted in an amount of virus particles that was suitable to be counted in the plaque assay. A large number of plaques (>50) was found when Vero E6 cells were incubated with a mixture of SARS-CoV-2 and a medium control (Figure 1, center wells). When the same amount of virus particles was either incubated with ViruProX^®^ (Figure 1, upper lane, left well) or BacterX^®^ pro (Figure 1, lower lane, left well) no plaques were found, indicating that both mouth rinsing solutions were able to reduce the 1 × 10^6^ pfu below the detection limit (8 × 10^4^ pfu/mL). When mouth rinsing solutions were incubated with Vero E6 cells alone in the same concentration as in the mixture with SARS-CoV-2, no plaques or cell destruction was found, indicating that the mouth rinsing solutions showed no cytotoxic effect (Figure 1, upper and lower lane, right well).

### 3.2. Identification of Effective Compounds

To identify the effective chemical compound of the two mouth rinsing solutions, 0.05% CPC, 0.1% CHX, 1.5% H_2_O_2_, and a combination of 0.1% CHX and 0.05% CPC were investigated for their virucidal potential using the same assay setup as described above. A virucidal activity could be observed for 0.05% CPC, even though few plaques were found (Figure 2, 1st lane, left well). No reduction of viral plaques, as compared to the medium control, was found when virus particles were treated with either 1.5% H_2_O_2_, a component of ViruProX^®^, or 0.1% CHX solutions, a component of BacterX^®^ pro (Figure 2, 2nd and 3rd lane, left well). These results indicate that the 0.05% CPC solution, which is present in both mouth rinsing solutions, is responsible for the virucidal effect against SARS-CoV-2. Interestingly, when a combination of 0.1% CHX and 0.05% CPC was tested for its virucidal potential, the number of plaques was strongly but not completely reduced compared to the medium control (Figure 2, 4th lane, left and center well).

### 3.3. Quantification of the Virucidal Effect

For further analysis and quantification of the results, two methods were used. First, the plaques were counted in each well, and the virus titer was determined. Compared to the medium control, ViruProX^®^ and BacterX^®^ pro significantly reduced the virus titer below the detection limit of 8 × 10^4^ pfu/mL (Figure 3A,B). Incubation with ViruProX^®^ resulted in a reduction of the virus titer by ≥ 6.8 × 10^6^ pfu/mL (≥1.9 log_10_ fold) compared to the medium control, while BacterX^®^ pro led to a reduction by ≥8.4 × 10^6^ pfu/mL (≥2.0 log_10_ fold). The combination of 0.1% CHX with 0.05% CPC (Figure 3C) and 0.05% CPC (Figure 3D) also reduced the virus titer significantly (6.7 × 10^6^ pfu/mL and 5.6 × 10^6^ pfu/mL/1.2 log_10_ fold and 0.7 log_10_ fold). While 0.1% CHX (Figure 3E) and 1.5% H_2_O_2_ (Figure 3F) showed no reduction in the virus titer. Furthermore, the crystal violet staining was dissolved with methanol, the absorbance of each sample was measured at 595 nm and compared to the no-virus control and the medium control. High absorbance values indicate an intact cell layer. Plaque formation (cytopathic effect) reduces the number of cells in the well, therefore less crystal violet is present, which is represented by a lower absorbance value at 595 nm. ViruProX^®^ (Figure 3G), BacterX^®^ pro (Figure 3H), the combination of 0.1% CHX with 0.05% CPC (Figure 3I), 0.05% CPC (Figure 3J), and 0.1% CHX (Figure 3K) showed a significant reduction of the cytopathic effect, compared to the medium control. In contrast, 1.5% H_2_O_2_ had no effect (Figure 3L).

## 4. Discussion

As of February 2021, the SARS-CoV-2 pandemic still controls most areas of our daily life. Transmission of SARS-CoV-2 occurs via direct contact with respiratory aerosols or droplets from infected individuals that result from sneezing, coughing, or talking [4,23]. Because of the fact that the oral cavity harbors high viral loads and the pharynx is thought to serve as the main site of viral replication, efforts are made to prevent transmission of the virus primarily by wearing masks [2,24,25,26]. Dental treatment often generates high amounts of aerosols and additional precautions must be taken due to the close contact with patients [27]. Therefore, the use of preprocedural mouth rinses, due to their antimicrobial effects, is more important than ever. It can be assumed that mouth rinsing can reduce the number of infectious viral particles and therefore the risk of transmission [3,8]. Previous research studies on SARS-CoV-2-related viruses (e.g., SARS-CoV and MERS-CoV) showed that solutions containing CHX, CPC, and H_2_O_2_ can indeed reduce viral load [26].

The present study demonstrates that two mouth rinsing solutions, ViruProX^®^ and BacterX^®^ pro successfully inactivate SARS-CoV-2 below the detection limit (reduction by ≥ 1.9 and ≥ 2.0 log_10_ fold) (Figure 3A,B). The mouth rinsing solution ViruProX^®^ contains 0.05% cetylpyridinium chloride (CPC) and 1.5% hydrogen peroxide (H_2_O_2_) and BacterX^®^ pro contains 0.1% chlorhexidine (CHX), 0.05% CPC, and 0.005% fluoride (F^-^). BacterX^®^ is also available with ethanol but, as ethanol may have a virucidal effect per se, the version without ethanol was tested here. When testing 0.05% CPC, 0.1% CHX, and 1.5% H_2_O_2_ alone, only 0.05% CPC led to a significant plaque reduction compared to the medium control (reduction by 0.7 log_10_ fold) (Figure 3D–F). Thus, the null hypothesis of this study was confirmed for 0.1% CHX and 1.5% H_2_O_2_, which showed no significant difference in the titer compared to the medium control. However, the null hypothesis was rejected for 0.05% CPC, the combination of 0.1% CHX and 0.05% CPC, ViruProX^®^, and BacterX^®^ pro which showed a significant difference in the titer and the cytopathic effect compared to the medium control. Therefore, it is likely that most of the virucidal activity of the two mouth rinse solutions results from CPC. Interestingly the combination of 0.1% CHX and 0.05% CPC led to a stronger reduction in plaques (1.2 log_10_ fold) than 0.05% CPC alone, although 0.1% CHX alone did not reduce the viral titer significantly (Figure 2 and Figure 3), meaning 0.1% CHX and 0.05% CPC in combination might act synergistically. It can be assumed that the same applies for other components of ViruProX^®^ and BacterX^®^ pro as the virucidal activity against SARS-CoV-2 was stronger compared to CPC or the combination of CHX and CPC (Figure 2 and Figure 3). However, the pure combination of H_2_O_2_ and CPC, as it was done for CPC and CHX, and also sodium fluoride alone, which is included in BacterX^®^ pro, was not tested. A limitation of the assay is the relatively high detection limit of 8 × 10^4^ pfu/mL, meaning the detection of zero plaques for ViruProX^®^ and BacterX^®^ pro could either represent the complete inactivation of SARS-CoV-2 or only a reduction below the assay’s detection limit. In reference to the clinical situation a limitation of the study might be seen in the sample preparation as it does not fully reflect the situation in the oral cavity. Saliva is reported to be a viral reservoir and carries median viral loads of 10^5^ genome copies/mL [4,28]. The virus titer of the samples used in the present study was 1.25 × 10^7^ pfu/mL and can therefore be considered representative for the viral load in the oral cavity. Furthermore, saliva in the mouth will dilute the mouth rinsing solution during the rinsing procedure, with a flow rate of approximately 5 mL/min, during the rinsing procedure of 30 s, 2.5 mL of saliva will accumulate in the mouth [29]. Using 10 mL of rinsing solution this would correspond to a dilution factor of 0.25. In the present study the mouth rinsing solution was diluted 1:2 with the viral sample. With that the dilution factor exceeds the dilution occurring in the mouth. Therefore, the virucidal activity measured in our assay might even underrate the antiviral effect in the in vivo situation.

Similar studies investigating antiviral properties of mouth rinse solutions have been performed previously. Bidra et al. [7] investigated the virucidal effect of H_2_O_2_ against SARS-CoV-2. They showed a minimal virucidal effect for 3% H_2_O_2_ and 6% H_2_O_2_ in a 1:2 dilution with the virus for 30 s, comparable to our setup, which resulted in a reduction of the viral titer by 1.0 to 1.8 log_10_ fold compared to the virus control [7]. Another study tested, among others, 0.2% CHX and 1.5% H_2_O_2_ and 0.015% dequalinium chloride, a compound related to CPC in mouth rinses. The concentrations used were comparable to the present study. Consistent with our results, they observed a strong virucidal effect for dequalinium chloride (reduction by ≥2.6 to ≥3.11 log_10_ fold) and they also found a small reduction for CHX (reduction by 0.78 to 1.17 log_10_ fold) and H_2_O_2_ (reduction by 0.33 to 0.78 log_10_ fold) [9]. An explanation for their results for CHX and H_2_O_2_ compared to our study might be found in the different SARS-CoV-2 isolates used. As Meister et al. [9] observed, different isolates can vary in their sensitivities to CHX and H_2_O_2_ [9]. Furthermore, a clinical trial has been performed to evaluate the in vivo efficacy of CPC and CHX to reduce the SARS-CoV-2 viral titer in the salivary of infected patients 5 min, 3 h, and 6 h post rinse. Although the patient number per group was small (*n* = 2–6) they could show a significant reduction in the viral titer (relative fold change of Ct value) for CPC 5 min and 6 h post rinse. No significance was reached for the CHX group compared to the water control [14]. The results support our findings and demonstrate that mouth rinsing with solutions containing CPC can successfully reduce the SARS-CoV-2 viral load in the oral cavity. The strong effect seen for CPC could be due to its positively charged surface-active components which may destroy the viral envelope.

The present study demonstrates the virucidal activity against SARS-CoV-2 for both mouth rinses ViruProX^®^ and BacterX^®^ pro. Both successfully inactivated 1 × 10^6^ pfu SARS-CoV-2. Furthermore, after testing the individual components, it could be shown that 0.05% CPC was able to strongly reduce infectious SARS-CoV-2 particles within 30 s incubation. Therefore, it can be hypothesized that CPC, which is present in both mouth rinses, is the effective component and we would recommend the use of mouth rinses containing CPC, like ViruProX^®^ and BacterX^®^ pro, in preprocedural use in dental practice to reduce the risk of transmission of SARS-CoV-2 during dental treatments. Another possible indication outside the dental practice could be rinsing and gurgling of the mouth with BacterX^®^ pro in order reduce the total load of SARS-CoV-2 after a possible encounter. BacterX^®^ pro is in contrast to ViruProX^®^ a cosmetic product according to EU regulations and may be used at home, too [30]. Clinical studies evaluating the virucidal efficacy of ViruProx^®^ and BacterX^®^ pro would need to be conducted to further support our findings and determine the duration of viral reduction in the oral cavity to ensure they exceed the duration for standard dental procedures. In regard to the results presented here, an update of guidelines recommending preprocedural mouth rinsing with solutions based on H_2_O_2_ or CHX might be advisable.

## Figures and Tables

**Figure 1 microorganisms-09-00521-f001:**
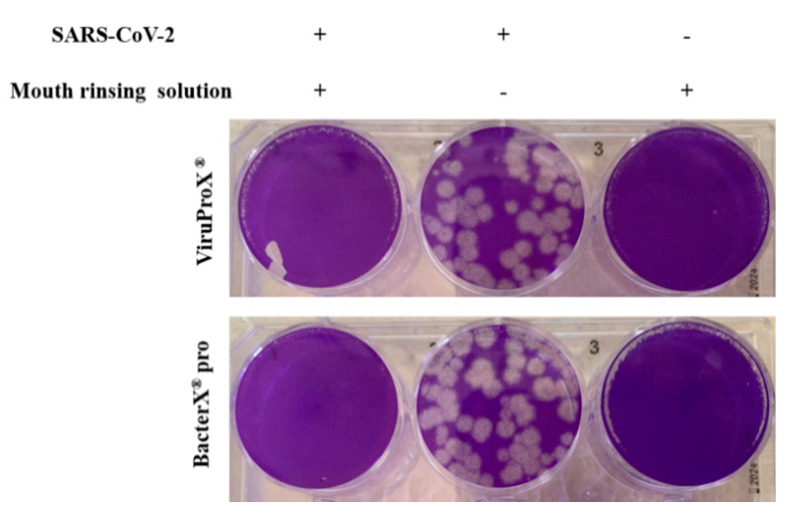
Reduced infectivity of SARS-CoV-2 after incubation with mouth rinsing solutions. SARS-CoV-2 (1 × 10^6^ pfu) was incubated for 30 s with two mouth rinsing solutions ViruProX^®^ and BacterX^®^ pro (left well) and tested for infectious viral particles in a standard plaque assay on Vero E6 cells, compared to a medium control (center well) and a no-virus control (right well). Each solution was tested in duplicates in two independent experiments. ViruProX^®^ and BacterX^®^ pro lead to a complete reduction of infectious viral particles.

**Figure 2 microorganisms-09-00521-f002:**
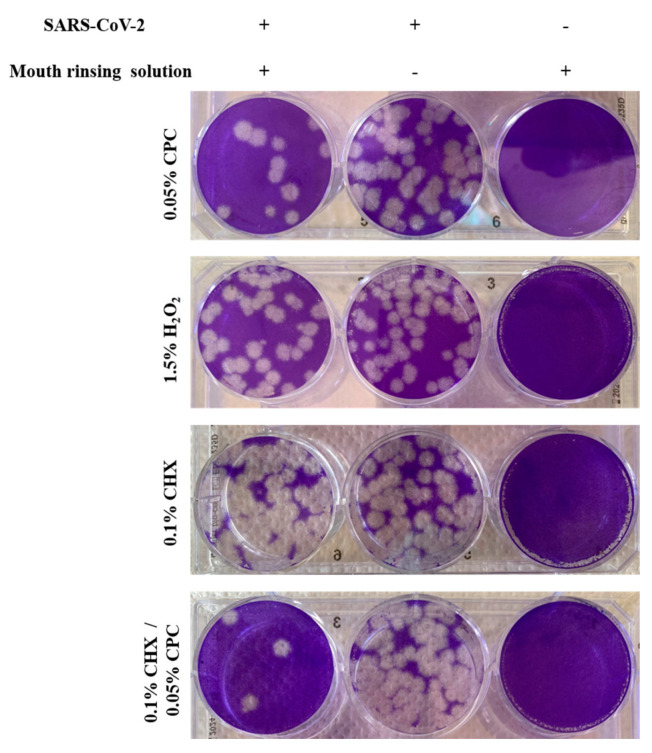
Virucidal effect of the individual components of the two mouth rinsing solutions against SARS-CoV-2. The individual components of ViruProX^®^ and BacterX^®^ pro, 0.05% cetylpyridinium chloride (CPC), 1.5% hydrogen peroxide (H_2_O_2_), 0.1% chlorhexidine (CHX), and the combination of 0.1% CHX with 0.05% CPC, were incubated with SARS-CoV-2 (1 × 10^6^ pfu) for 30 s and tested for infectious viral particles in a standard plaque assay on Vero E6 cells (left well), compared to a medium control (center well) and a no-virus control (right well). Each solution was tested in duplicates in two independent experiments. A reduction resulted from 0.05% CPC and the combination of 0.1% CHX with 0.05% CPC, while no virucidal effect was observed for 0.1% CHX and 1.5% H_2_O_2_.

**Figure 3 microorganisms-09-00521-f003:**
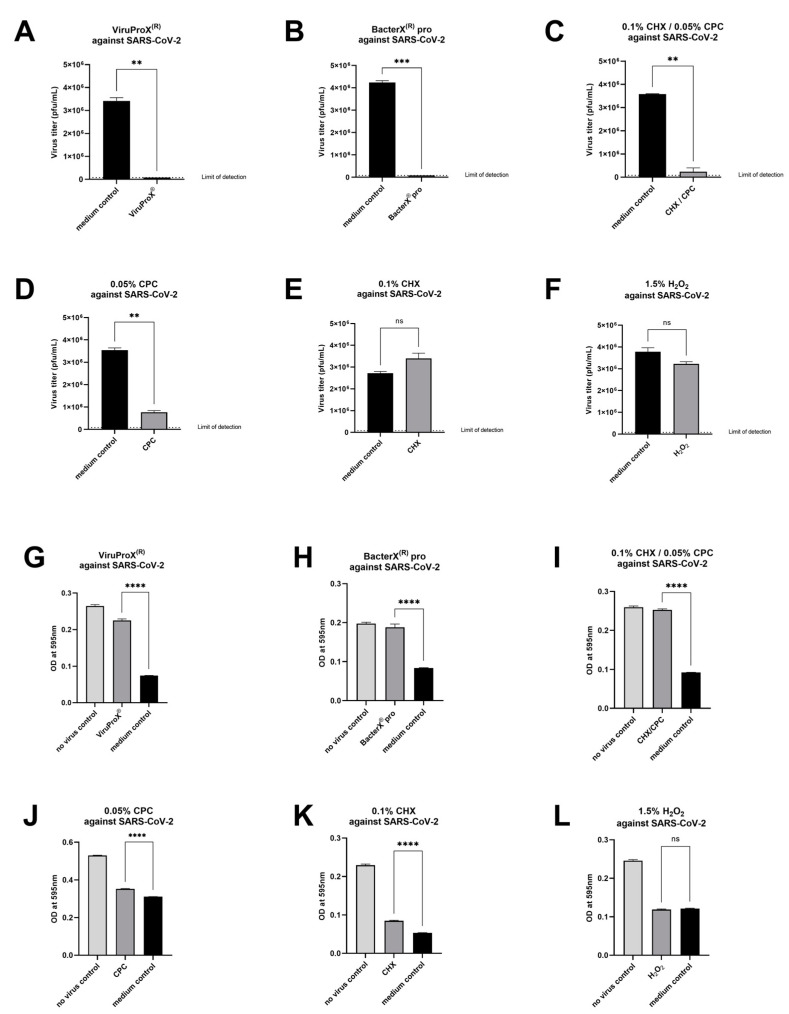
Determination of the virus titer and quantification of the cytopathic effect. The viral titer (pfu/mL) of the samples was determined by counting the plaques in each well (*n* = 1–4). Statistical analysis was performed using unpaired t-test (*p* > 0.05 (ns), *p* ≤ 0.01 (**), *p* ≤ 0.001 (***), *p* ≤ 0.0001 (****)) (**A**–**F**). The crystal violet staining of the cell layers was dissolved with methanol and the absorbance of the samples was measured at 595 nm. Staining of the no-virus control represents the intact cell layer and the medium control the maximal cytopathic effect. Statistical analysis was performed using one-way ANOVA with Tukey’s multiple comparisons test (*p* > 0.05 (ns), *p* ≤ 0.01 (**), *p* ≤ 0.001 (***), *p* ≤ 0.0001 (****)) (**G**–**L**).

## Data Availability

The data presented in this study are contained within the article and Appendix A.

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
