# Peer review of "Inactivation of SARS-CoV-2 through Treatment with the Mouth Rinsing Solutions ViruProX® and BacterX® Pro"

_microorganisms, 2021, doi:10.3390/microorganisms9030521_

Round 1

Reviewer 1 Report

The study is of great interest and addresses and extremely timely issue. Previous literature reported on the use of pre-procedural mouth rinses in dental practice, please further address this fact in the introduction.

Study limitations should be further discussed, especially in the light of the fact that saliva has been reported to be a viral reservoir.

Author Response

Response to Reviewer 1

Line references refer to manuscript version in track mode

  • The study is of great interest and addresses and extremely timely issue. Previous literature reported on the use of pre-procedural mouth rinses in dental practice, please further address this fact in the introduction.

First, we would like to thank the reviewer for appreciating our study. We have addressed the use of pre-procedural mouth rinses in dental practice in the introduction and added a summary of recommendations by WHO and CDC, because these are more current compared to standard literature. The information is now supplemented with the following paragraph (lines 103-107):

“Official recommendations for dental practice therefore give little guidance on specific compounds for pre-procedural mouth rinses. Recommendations by the WHO include the use of 1% H2O2 and 0.2% PI solutions. The CDC lists, CHX, essential oils, PI and CPC to be used in pre-procedural mouth rinses.”

  • Study limitations should be further discussed, especially in the light of the fact that saliva has been reported to be a viral reservoir.

The limitations of the study are now further discussed (lines 557-568):

In reference to the clinical situation a limitation of the study might be seen in the sample preparation as it does not fully reflect the situation in the oral cavity. Saliva is reported to be a viral reservoir and carries median viral loads of 105 genome copies /mL (Li et al.+ To et al, 2020). The virus titer of the samples used in the present study was 1.25x107 pfu/mL and can therefore be considered representative for the viral load in the oral cavity.

Furthermore, saliva in the mouth will dilute the mouth rinsing solution during the rinsing procedure, with a flow rate of approximately 5 mL/min, during the rinsing procedure of 30 seconds, 2.5 mL of saliva will accumulate in the mouth (Iorgulescu 2009). Using 10 mL of rinsing solution this would correspond to a dilution factor of 0.25. In the present study the mouth rinsing solution was diluted 1:2 with the viral sample. With that the dilution factor exceeds the dilution occurring in the mouth. Therefore, the virucidal activity measured in our assay might even underrate the antiviral effect in the in vivo situation.”

Reviewer 2 Report

This research is under the scope of this journal; the topic is relevant for readers, and this research deals with potentially significant knowledge to the field.

However, there are some concerns in the about the present manuscript:

Abstract

  • How many samples? Identified in the abstract.
  • The authors should describe how the results were expressed and statistical analysis performed. 
  • In the results, is important to show more information, add some of the p-values.

Introduction

  • What is the importance of this study for clinical? What is the gap in the literature? Is necessary to change some procedures on the dental office? Which results are comparable with others study in oral care?
  • Page 1 line 42 “Therefore, specific infection control measures have been implemented 42 in dental offices” recommend a reference to support (COVID19-NOC | PDF - Rapid Guidelines on Covid 19 for Dentistry - based in National Institute for Health and Care Excellence (www.NICE.org) ISBN-13 (15) 978-989-26-2072- 5 doi https://doi.org/10.14195/978-989-26-2072-5;http://monographs.uc.pt/iuc/catalog/view/28/343/627-1)
  • Add more information to these to products ViruProX and Bacter X (Manufacturer, City, Country), Add a supplementary material on the table with composition, also the lot, date used in this study. Also, the indication of the manufacturer`s.
  • What was the null hypothesis for this study? Add on the last sentence in the introduction and reject in the discussion.

Materials and Methods

  • Please include a statement in the Material and Methods section that the study has been approved by the institutional ethics committee and provide the number of the process.
  • Made a flowchart, to explain to reads the sequence of the study and N.
  • How was the sample calculated? Did the authors perform a power analysis to evaluate if this sample size was appropriate?
  • When mentioning materials or devices: for some of them you don't mention the manufacturer at all, for some you mention only the manufacturer, for some the manufacturer and city, for some you mention the manufacturer and city/ country.
  • Test solution: “Dr Wittmann GmbH & Co KG, 64673, Zwingenberg (Germany) and E.M.S. Electro Medical Systems S.A.,1260 Nyon (Switzerland)” these information’s must move to an Acknowledgments.
  • “1:40,000” change for 1:40.000. Correct this for numeric form in all manuscript.

Results

  •  
  • The use of personal pronouns should be avoided. Example “We investigated”.

Discussion 

  • “Bidra and colleagues” correct to Bidra et al. (reference) ….when identified the author change also in all document for other references.
  • Please, clarified other limitations of this study?
  • And, clarified the future perspectives.

References

  • References are problems with the insertion on the text “Error! Reference source not found”, Please correct that.
  • But references are not standardized. The titles of references have a different format, 
    the title of the article is written in capital letters at the beginning of words, others only in lower case. Also, the standardized format of presentation in the journal's name. Because names have written in a different format, one is not abbreviated, others are not.
  • The reference is on the final of the sentence. But, when you had in the text the “authors et al.” references should come immediately afterwards, not in the final of the sentence. (corrected in all manuscript).

Author Response

Response to Reviewer 2 

Line references refer to manuscript version in track mode 

Abstract 

  1. How many samples? Identified in the abstract.  

The authors should describe how the results were expressed and statistical analysis performed. In the results, is important to show more information, add some of the p-values. 

We would like to apologize for not giving this information. To address this, we added a quantitative analysis of our study. The statistical analysis was performed using unpaired t-test (p<0.05) for the virus titer analysis and by using a one-way ANOVA with Tukey's multiple comparisons test (p<0.05) for quantification of the cytopathic effect. We have added this information in the legend of Figure 3 and in the method section (lines 270-275).  

Introduction 

2. What is the importance of this study for clinical? What is the gap in the literature? Is necessary to change some procedures on the dental office? Which results are comparable with others study in oral care? Page 1 line 42 “Therefore, specific infection control measures have been implemented 42 in dental offices” recommend a reference to support (COVID19-NOC | PDF - Rapid Guidelines on Covid 19 for Dentistry - based in National Institute for Health and Care Excellence (www.NICE.org) ISBN-13 (15) 978-989-26-2072- 5 doi https://doi.org/10.14195/978-989-26-2072-5;http://monographs.uc.pt/iuc/catalog/view/28/343/627-1) 

With our study we intend to provide information on mouth rinse solutions to be used pre-procedurally in dental offices, to reduce the risk of SARS-CoV-2 transmission. 

To date several studies indicate a virucidal effect for cetylpyridinium chloride and povidone iodine against SARS-CoV-2. But there is a limited number of mouth rinses “ready to use” that are recommended for SARS-CoV-2 transmission prevention. Here, we demonstrate that the two “ready to use” mouth rinsing solutions BacterX pro and ViruProX, both containing cetylpyridinium chloride, successfully inactivate infectious SARS-CoV-2 viral particles in vitro. Further analysis revealed that CPC is the active compound and that CHX and H2O2 did not show virucidal activity against SARS-CoV-2. Our findings are in agreement with studies conducted by Statkute et al. 2020 and Bidra et al. 2020. Thank you for your recommendation concerning COVID-19 guidelines in dentistry. The guideline recommends rinsing of the mouth and gargling with antimicrobial solutions having the ability to oxidize this type of virus (SARS-CoV-2) e.g. 1% hydrogen peroxide or 0.2% povidone-iodine 30 to 60 seconds before starting consultation. 

Based on our study a change in this guideline might be advisable in terms of replacing the recommendation to use hydrogen peroxide solutions by solutions containing cetylpyridinium chloride, as hydrogen peroxide had no significant virucidal effect in our study. 

3. Add more information to these two products ViruProX and Bacter X (Manufacturer, City, Country), Add a supplementary material on the table with composition, also the lot, date used in this study. Also, the indication of the manufacturer`s. 

The information of ViruProX and Bacter X pro is already given in the section 2.2. All test solutions were provided by Dr. Wittmann GmbH & Co KG, 64673, Zwingenberg (Germany) and E.M.S. Electro Medical Systems S.A.,1260 Nyon (Switzerland). This information is now shifted to the acknowledgment (lines 812-814) and the following table was added as supplementary. 

Test solution 

Composition 

Manufacturer 

Lot 

Recommendation of the manufacturer 

ViruProX®  

Aqua, Propylene Glycol, Glycerin, PEG-40, Hydrogenated Castor Oil, Aroma, Hydrogen Peroxide (1.5%), Cetylpyridinium Chloride (0.05%), Sucralose, Erythritol, Sodium Saccharin 

Dr. Wittmann GmbH & Co KG 

Lot.: 2005191 

gargle and rinse with at least 10mL undiluted solution for 40 s  

BacterX® pro 

Aqua, Glycerol, Propylene Glycol, PEG-40, Hydrogenated Castor Oil, Sucralose, Chlorhexidine Digluconate (0.1%), Cetylpyridinium Chloride (0.05%), Xylite, Aroma, Sodium Fluoride (0.005%), CI 42051  

Dr. Wittmann GmbH & Co KG 

Lot.: 2006221 

rinse the mouth with undiluted solution for 20-30s 

0.05% CPC 

Dr. Wittmann GmbH & Co KG 

Lot: K93839740945 

Preparation Date: 14.07.2020 

0.1% CHX 

Dr. Wittmann GmbH & Co KG 

Lot.: 3-8962-9-02-17,  

Preparation Date: 14.07.2020 

1.5% H2O2 

Dr. Wittmann GmbH & Co KG 

1200203404, Preparation Date: 14.07.2020 

4. What was the null hypothesis for this study? Add on the last sentence in the introduction and reject in the discussion. 

We have made the changes according to the reviewer’s suggestion. The definition of the null hypothesis was integrated as the last sentence in the introduction (lines 116-119): 

As null hypothesis of the study it was assumed that none of the tested solutions has a virucidal activity against SARS-CoV-2 and therefore will show no significant difference compared to the medium control. 

The rejection of the null hypothesis is now mentioned in the discussion (lines 540-545): 

The null hypothesis of this study was confirmed for 0.1% CHX and 1.5% H2O2, which showed no significant difference in the titer compared to the medium control. However, the null hypothesis was rejected for 0.05% CPC, the combination of 0.1% CHX and 0.05% CPC, ViruProX and BacterX pro which showed a significant difference in the titer and the cytopathic effect compared to the medium control.  

Materials and Methods 

5. Please include a statement in the Material and Methods section that the study has been approved by the institutional ethics committee and provide the number of the process. Made a flowchart, to explain to reads the sequence of the study and N. How was the sample calculated? Did the authors perform a power analysis to evaluate if this sample size was appropriate?  

Since this was an in vitro study conducted in Vero E6 cells and not a clinical study, there was no need to approve this study by the ethics committee. A flowchart, as used in clinical trials, was consequently also not suitable in the present study. The same applies to a power analysis.  

6. When mentioning materials or devices: for some of them you don't mention the manufacturer at all, for some you mention only the manufacturer, for some the manufacturer and city, for some you mention the manufacturer and city/ country. 

We have to apologize for this inconsistency and would like to thank the reviewer for this comment. We have revised the complete manuscript accordingly.  

Test solution: “Dr Wittmann GmbH & Co KG, 64673, Zwingenberg (Germany) and E.M.S. Electro Medical Systems S.A.,1260 Nyon (Switzerland)” these information’s must move to an Acknowledgments. This information is moved to the Acknowledgements. 

7. “1:40,000” change for 1:40.000. Correct this for numeric form in all manuscript. 

Again, we must apologize for this negligence. The numeric form is changed in all parts of the manuscript.  

Results 

8. The use of personal pronouns should be avoided. Example “We investigated”. 

We apologize for the inconvenience and corrected the respective passages in the manuscript. 

Discussion  

9. Bidra and colleagues” correct to Bidra et al. (reference) ….when identified the author change also in all document for other references. 

The reference output is changed in the complete manuscript.  

10. Please, clarified other limitations of this study? And, clarified the future perspectives. 

The limitations of the study are now further discussed: (lines 557-568) 

In reference to the clinical situation a limitation of the study might be seen in the sample preparation as it does not fully reflect the situation in the oral cavity. Saliva is reported to be a viral reservoir and carries median viral loads of 105 genome copies /mL (Li et al.+ To et al, 2020). The virus titer of the samples used in the present study was 1.25x107 pfu/mL and can therefore be considered representative for the viral load in the oral cavity.  

Furthermore, saliva in the mouth will dilute the mouth rinsing solution during the rinsing procedure, with a flow rate of approximately 5 mL/min, during the rinsing procedure of 30 seconds, 2.5 mL of saliva will accumulate in the mouth (Iorgulescu 2009). Using 10 mL of rinsing solution this would correspond to a dilution factor of 0.25. In the present study the mouth rinsing solution was diluted 1:2 with the viral sample. With that the dilution factor exceeds the dilution occurring in the mouth. Therefore, the virucidal activity measured in our assay might even underrate the antiviral effect in the in vivo situation. 

Future perspectives are now given in the discussion (lines 755-757): 

In regard to the results presented here, an update of guidelines recommending pre-procedural mouth rinsing with solutions based on H2O2 or CHX might be advisable. 

References 

11. References are problems with the insertion on the text “Error! Reference source not found”, Please correct that. But references are not standardized. The titles of references have a different format, The title of the article is written in capital letters at the beginning of words, others only in lower case. Also, the standardized format of presentation in the journal's name. Because names have written in a different format, one is not abbreviated, others are not. The reference is on the final of the sentence. But, when you had in the text the “authors et al.” references should come immediately afterwards, not in the final of the sentence. (corrected in all manuscript). 

The references are corrected in the complete manuscript 

Round 2

Reviewer 2 Report

This research is under the scope of this journal; the topic is interesting for readers and this research deals with potentially significant knowledge to the field and an open new way for future studies.

The authors improved the quality of the manuscript after the reviewer's indications. 

But still, need to correct minor correction: Based on the authors' response, it seems to be relevant to include in the discussion two points.

(discussion) The authors refer that based on the current guidelines (COVID-19 guidelines in dentistry (COVID19-NOC | PDF - Rapid Guidelines on Covid 19 for Dentistry - based in National Institute for Health and Care Excellence (www.NICE.org) ISBN-13 (15) 978-989-26-2072- 5 doi https://doi.org/10.14195/978-989-26-2072-5;http://monographs.uc.pt/iuc/catalog/view/28/343/627-1) 

"The guideline recommends rinsing of the mouth and gargling with antimicrobial solutions having the ability to oxidize this type of virus (SARS-CoV-2) e.g. 1% hydrogen peroxide or 0.2% povidone-iodine 30 to 60 seconds before starting the consultation. "

And

To date, several studies indicate a virucidal effect for cetylpyridinium chloride and povidone-iodine against SARS-CoV-2. But there is a limited number of mouth rinses “ready to use” that is recommended for SARS-CoV-2 transmission prevention. Here, we demonstrate that the two “ready to use” mouth rinsing solutions BacterX pro and ViruProX, both containing cetylpyridinium chloride, successfully inactivate infectious SARS-CoV-2 viral particles in vitro."

For this reason,  this study may contribute to change the guidelines. 

(discussion) An advantage in the flavour of the new solutions(commercials), compared to the other solutions .... since it is an important aspect for consumers! The flavour of H2O2 is horrible. The flavour is an important aspect for consumers!

Author Response

Response to Reviewer 2 (Round 2)

This research is under the scope of this journal; the topic is interesting for readers and this research deals with potentially significant knowledge to the field and an open new way for future studies. The authors improved the quality of the manuscript after the reviewer's indications. But still, need to correct minor correction: Based on the authors' response, it seems to be relevant to include in the discussion two points.

Discussion

  1. The authors refer that based on the current guidelines (COVID-19 guidelines in dentistry (COVID19-NOC | PDF - Rapid Guidelines on Covid 19 for Dentistry - based in National Institute for Health and Care Excellence (NICE.org)

ISBN-13 (15) 978-989-26-2072- 5 doi https://doi.org/10.14195/978-989-26-2072-5; http://monographs.uc.pt/iuc/catalog/view/28/343/627-1) 

"The guideline recommends rinsing of the mouth and gargling with antimicrobial solutions having the ability to oxidize this type of virus (SARS-CoV-2) e.g. 1% hydrogen peroxide or 0.2% povidone-iodine 30 to 60 seconds before starting the consultation. "

Thank you for your comment, we already addressed this point in the manuscript by citing recommendations of the WHO and CDC. (lines 64-68) and by adding the following sentence at the end of the discussion: “In regard to the results presented here, an update of guidelines recommending pre-procedural mouth rinsing with solutions based on H2O2 or CHX might be advisable.” (lines 321-323).

  1. To date, several studies indicate a virucidal effect for cetylpyridinium chloride and povidone-iodine against SARS-CoV-2. But there is a limited number of mouth rinses “ready to use” that is recommended for SARS-CoV-2 transmission prevention. Here, we demonstrate that the two “ready to use” mouth rinsing solutions BacterX pro and ViruProX, both containing cetylpyridinium chloride, successfully inactivate infectious SARS-CoV-2 viral particles in vitro."

For this reason, this study may contribute to change the guidelines. 

An advantage in the flavour of the new solutions(commercials), compared to the other solutions .... since it is an important aspect for consumers! The flavour of H2O2 is horrible. The flavour is an important aspect for consumers!

From a marketing perspective you are right, and the flavor of a mouth rinsing solution would be very important in this regard. Our intention with this publication is purely scientific. Therefore, the efficacy of the solutions to inactivate SARS-CoV-2 is most important to us. Nevertheless, we demonstrated the virucidal activity against SARS-CoV-2 for two mouth rinsing solution, one with H2O2, one without. This means consumers could choose which mouth rinse to use depending on their taste.